# The "Periodic Table" of 1-methylbenzotriazole: Zinc(II) Complexes

Christina Stamou [1], Eleftheria Barouni [1], John C. Plakatouras [2], Michael M. Sigalas [3], Catherine P. Raptopoulou [4], Vassilis Psycharis [4,*], Evangelos G. Bakalbassis [5,*] and Spyros P. Perlepes [1,6,*]

[1] Department of Chemistry, University of Patras, 26504 Patras, Greece; xrstamou@gmail.com (C.S.); riabarou@gmail.com (E.B.)
[2] Department of Chemistry, University of Ioannina, 45110 Ioannina, Greece; iplakatu@uoi.gr
[3] Department of Materials Science, University of Patras, 26504 Patras, Greece; sigalas@upatras.gr
[4] Institute of Nanoscience and Nanotechnology, NCSR "Demokritos", 15310 Aghia Paraskevi Attikis, Greece; c.raptopoulou@inn.demokritos.gr
[5] Department of Chemistry, Aristotle University of Thessaloniki, University Campus, 54124 Thessaloniki, Greece
[6] Institute of Chemical Engineering Sciences, Foundation for Research and Technology (FORTH/ICE-HT), Platani, P.O. Box 1414, 26504 Patras, Greece
* Correspondence: v. psycharis@inn.demokritos.gr (V.P.); bakalbas@chem.auth.gr (E.G.B.); perlepes@upatras.gr (S.P.P.)

**Abstract:** In an attempt to fill in the empty Zn position in the "Periodic Table" of 1-methylbenzotriazole (Mebta), reactions between Zn(II) sources and this ligand were carried out. The detailed synthetic studies provided access to complexes [ZnX$_2$(Mebta)$_2$] (X = Cl, **1**; X = Br, **3**; X = I, **4**), (MebtaH)$_2$[ZnCl$_4$] (**2**), tet-[Zn(NO$_3$)$_2$(Mebta)$_2$] (**5**), oct-[Zn(NO$_3$)$_2$(Mebta)$_2$] (**6**), and [Zn(Mebta)$_4$](Y)$_2$ [Y = ClO$_4$, **7**; Y = PF$_6$, **8**]. Solid-state thermal decomposition of **2** leads to **1** in quantitative yield. The structures of **3**, **4**, **5**, **6**, and **7** were determined by single-crystal crystallography. The structures of the remaining complexes were proposed based on spectroscopic evidence. In all compounds, Mebta behaves as monodentate ligand using the nitrogen of the position 3 as donor. Complexes **1**–**4**, **7,** and **8** are tetrahedral. Complexes **5** and **6** are isostoichiometric and their preparation in pure forms depends on the reaction conditions; in the former the Zn$^{II}$ atom has a tetrahedral geometry, whereas in the latter the metal ion is octahedral. This case of rare isomerism arises from the monodentate (in **5**) vs. bidentate (in **6**) coordination of the nitrato groups. Extensive π–π stacking interactions and non-classical H bonds build interesting 3D architectures in the structurally characterized complexes. The compounds were characterized by IR, far-IR, and Raman spectroscopies in the solid state, and the data were interpreted in terms of the structures (known or proposed) of the complexes and the coordination modes of the organic and inorganic ligands involved. The solid-state structures of the complexes are not retained in solution, as proven by NMR ($^1$H, $^{13}$C[$^1$H]) spectroscopy and molar conductivity data. The thermal decomposition study of **1** and **3** leads to stable intermediates with 1:1 stoichiometry, i.e., ZnX$_2$(Mebta). Based on far-IR spectra, polymeric tetrahedral structures are possible with simultaneous presence of terminal and bridging X$^-$ groups. Liquid-phase ab initio (MP2) and gas-phase DFT calculations, performed on Mebta and the nitrato complexes, respectively, shed light on the tendency of Mebta for N3-coordination, and the existence and relative stabilities of **5** and **6**.

**Keywords:** DFT studies; 1-methylbenzotriazole metal complexes; single-crystal X-ray studies; spectroscopic (IR, far-IR, Raman, NMR) studies; thermogravimetric analysis; zinc(II) complexes

## 1. Introduction

Azoles, e.g., pyrazoles and imidazoles, have written their own history in classical coordination chemistry in the last five decades [1]. A popular family of this class of ligands

are benzotriazoles (Figure 1; R, R′ = H or various non-donor and/or donor groups). Their basic skeleton consists of two fused five-(1,2,3-triazole) and six-(benzene) rings [2,3]. In the parent molecule (BtaH; R, R′ =H), the benzenoid form is more stable (by ~9 kcal mol$^{-1}$) than the quinonoid form, and thus most derivatives of BtaH appear as the N(1)-isomers [4]. The molecule is simultaneously a weak acid (pK$_a$ = 8.2) and a weak base [5].

R, R' = H (BtaH);
R = H, R' = Me, Cl, NO$_2$, COOH;
R = OH, R' = H;
R = CH$_2$COOH, R' = H etc.

Mebta

**Figure 1.** (**Left**) The general structural formula of BtaH and its derivatives; (**right**) 1-methylbenzotriazole (Mebta), the ligand used in this work.

Benzotriazoles are central "players" in organic [2,3,6], medicinal [7,8], environmental [9], physical and theoretical [4,10–14], and materials [15–22] chemistry from both the basic research and applications viewpoints. For example, in materials chemistry: (a) polar benzotriazoles (including BtaH) are used as corrosion inhibitors for metals (especially Cu) and their alloys, in antifreeze liquids, household detergents, aircraft de-icing fluids, and in the treatment of artifacts of historical and archaeological importance [15–17]; (b) BtaH and its amino-, azido-, and nitro-derivatives are being tested as energetic materials [18–20]; (c) *ortho*hydroxyphenyl benzotriazoles find extensive use as UV absorbers (stabilizers) in plastic materials, personal care products, and biosolids [21,22]; and (d) benzotriazole-decorated grapheme oxide is a very efficient material for removal of the uranyl ion from nuclear waste water [23].

Benzotriazole- and benzotriazolate-based ligands are also important in inorganic chemistry [24,25]. They have been used by several groups and our group in the synthesis of—amongst others—coordination clusters and cyclic cages [26–35], and coordination polymers [36–48], often with interesting magnetic, optical, and catalytic properties.

1-methylbenzotriazole (Mebta, Figure 1) is the archetype of simple N-substituted benzotriazoles. It is almost white and harmful to the respiratory system. Moreover, it is dangerous if ingested and causes irritation upon contact with eyes and skin. Recently, we performed a detailed study of the vibrational, electronic, and NMR spectra of Mebta, combined with Density Functional Theory (DFT) calculations employing various functional and basis sets [49]. The coordination chemistry of Mebta has been intensely studied by our group [50–53], and sporadically by other research teams [54,55]. A common characteristic in all studies is the monodentate ligation of Mebta through the N atom of position 3 (see Figure 1) of the azole ring. This exclusively monodentate coordination has been related to the inability of Mebta to act as corrosion inhibitor; for such a property, deprotonation and bridging behavior of the molecule is required [14,16,17]. Over the past three decades, we have been making an effort to build a "periodic table" of Mebta, studying its reactions with as many metal and metalloid ions as possible; one of our concerns is to study the possibility of N2, N3-bridging behavior for Mebta (such effort are unsuccessful so far). Regarding the first goal, we are now glad to fill the blank space of Zn(II) in this "periodic table" by describing the syntheses, structures, and spectroscopic characterization of the first Zn(II) complexes; we also support our experimental data by theoretical calculations, with emphasis on the investigation of the inability of Mebta to behave as a bidentate bridging ligand.

## 2. Results and Discussion

### 2.1. Synthetic Comments

For reasons outlined in "Introduction", we sought the preparation of Zn(II)/Mebta complexes. We started our efforts with zinc(II) halides. The 1:2 reactions between $ZnX_2$ and Mebta in EtOH gave complexes $[ZnX_2(Mebta)_2]$ (X = Cl, **1**; X = Br, **3**; X = I, **4**) in low to moderate yields, Equation (1). Changes of the molar ratio from 1:2 to 1:1 or 1:4 and the solvent from EtOH to MeCN led again to the isolation of **1–3** (analytical, IR, and Raman data); thus, no compounds of the empirical formulae $ZnX_2(Mebta)$ or $ZnX_2(Mebta)_4$ could be isolated, suggesting that the 1:2 complexes are the thermodynamically favored species under the conditions employed.

$$ZnX_2 + 2Mebta \xrightarrow{EtOH} [ZnX_2(Mebta)_2] \tag{1}$$
$$\mathbf{1, 3, 4}$$

Trying to prepare salts with protonated Mebta, i.e., $MebtaH^+$, we performed the 1:2 reaction between $ZnCl_2$ and Mebta in hydrochloric acid; slow evaporation of gaseous HCl from the reaction solution resulted in bis(1-methylbenzotriazolium)tetrachlorozincate, $(MebtaH)_2[ZnCl_4]$ (**2**), in yields depending on the final concentration of the system, Equation (2). Thermal decomposition of solid **2** at 205 °C under $N_2$ gave **1** in a quantitative yield, Equation (3).

$$ZnCl_2 + 2Mebta + 2HCl \xrightarrow{HCl/H_2O} (MebtaH)_2[ZnCl_4] \tag{2}$$
$$\mathbf{2}$$

$$(MebtaH)_2[ZnCl_4] \xrightarrow{N_2/205°C} [ZnCl_2(Mebta)_2] + 2HCl \uparrow \tag{3}$$
$$\mathbf{2} \qquad\qquad\qquad\qquad \mathbf{1}$$

Continuing our efforts, we used $Zn(NO_3)_2 \cdot 4H_2O$ as starting material. The possibility of bidentate (either chelating or bridging) behavior of $NO_3^-$ might give different chemistry compared with that of zinc(II) halides. Somewhat to our surprise, we isolated two mononuclear nitrato complexes with the same stoichiometry, but with different stereochemistry at the metal ion resulting from the different coordination mode of the nitrato ligands (monodentate vs. bidentate). The 1:2 reaction of $Zn(NO_3)_2 \cdot 4H_2O$ and Mebta in EtOH at room temperature led to the precipitation of tet-$[Zn(NO_3)_2(Mebta)_2]$ (**5**; the abbreviation "tet" denotes the tetrahedral geometry of $Zn^{II}$), Equation (4). The analogous reaction at 55 °C in the presence of excess of triethylorthoformate (TEOF) gave complex oct-$[Zn(NO_3)_2(Mebta)_2]$ (**6**; "oct" denotes the octahedral geometry of $Zn^{II}$), Equation (5). The preparative procedures are perfectly reproducible. These experimental observations indicate that the concentration of $H_2O$ in the reaction mixture affects the product identity. It is well known that TEOF is an efficient dehydrating agent in coordination chemistry (this capacity increases with increasing temperature) [55], Equation (6). It is hypothesized (no experimental evidence) that in the presence of $H_2O$ contained in EtOH, the octahedral species $[Zn(NO_3)_2(Mebta)_2(H_2O)_2]$ (with monodentate nitrates) is formed in solution, and the aqua ligands are removed during crystallization retaining the monodentate character of $NO_3^-$ and leading to **5**. When $H_2O$ is less (or practically absent) in the reaction mixture, the octahedral species $[Zn(NO_3)_2(Mebta)_2]$ (with bidentate nitrates) is formed in solution and this is precipitated, resulting in **6**. In both cases, it is assumed that EtOH can not compete with $H_2O$ or $NO_3^-$ for participation in the coordination sphere of $Zn^{II}$. Complexes **5** and **6** were also isolated using a 1:4 $Zn(NO_3)_2 \cdot 4H_2O$/Mebta reaction ratio.

$$Zn(NO_3)_2 \cdot 4H_2O + 2Mebta \xrightarrow{20°C/EtOH} tet-[Zn(NO_3)_2(Mebta)_2] + 4H_2O \tag{4}$$
$$\mathbf{5}$$

$$Zn(NO_3)_2 \cdot 4H_2O + 2Mebta + 4HC(OC_2H_5)_3 \xrightarrow{55^\circ C/EtOH} oct-[Zn(NO_3)_2(Mebta)_2] + \atop \textbf{6} \\ 4HCO(OC_2H_5)_3 + 8C_2H_5OH \tag{5}$$

$$HC(OC_2H_5)_3 + H_2O \xrightarrow{T} HCO(OC_2H_5) + 2C_2H_5OH \tag{6}$$

Our next step was to investigate if salts containing the $[Zn(Mebta)_4]^{2+}$ ion, i.e., cationic complexes, could be capable of existence. For this reason, we used $Zn(ClO_4)_2 \cdot 6H_2O$ as starting material because $ClO_4{}^-$ shows little tendency to coordinate to metal ions. From the 1:4 $Zn(ClO_4)_2 \cdot 6H_2O$/Mebta reaction solution in EtOH, the expected complex $[Zn(Mebta)_4](ClO_4)_2$ (**7**) was isolated in 45% yield after slow diffusion of $Et_2O$ into the reaction solution, Equation (7). Addition of an excess of $NH_4(PF_6)$ in the 1:4 $Zn(ClO_4)_2 \cdot 6H_2O$/Mebta reaction solution in EtOH (a metathesis reaction), followed by slow diffusion of n-hexane, resulted in the precipitation of $[Zn(Mebta)_4](PF_6)_2$ (**8**) in ~60% yield, Equation (8).

$$Zn(ClO_4)_2 \cdot 6H_2O + 4Mebta \xrightarrow{EtOH} [Zn(Mebta)_4](ClO_4)_2 + 6H_2O \atop \textbf{7} \tag{7}$$

$$Zn(ClO_4)_2 \cdot 6H_2O + 4Mebta + 2NH_4(PF_6) \xrightarrow{EtOH} [Zn(Mebta)_4](PF6)_2 + 2NH_4(ClO_4) + 6H_2O \atop \textbf{8} \tag{8}$$

### 2.2. Description of Structures

The structures of **3–7** were determined by single-crystal X-ray crystallography. Various structural plots are shown in Figures 2–8 and S1–S8. Selected distances and angles are listed in Tables 1–3.

**Table 1.** Selected bond lengths (Å) and angles (°) for complexes $[ZnBr_2(Mebta)_2]$ (**3**) and $[ZnI_2(Mebta)_2]$ (**4**).

| Bond Lengths (Å) and Angles (°) [a] | 3 | 4 |
|---|---|---|
| Zn-X1 | 2.367(1) | 2.554(1) |
| Zn-X2 | 2.353(7) | 2.567(1) |
| Zn-N3 | 2.034(3) | 2.050(4) |
| Zn-N13 | 2.070(4) | 2.042(4) |
| N1-N2 | 1.335(4) | 1.334(6) |
| N2-N3 | 1.321(4) | 1.315(7) |
| N11-N12 | 1.329(5) | 1.333(7) |
| N12-N13 | 1.323(4) | 1.321(7) |
| C9-N1 | 1.362(5) | 1.362(8) |
| C8-N3 | 1.384(5) | 1.367(7) |
| C10-N1 | 1.466(5) | 1.461(7) |
| X1-Zn-X2 | 116.4(3) | 115.4(1) |
| X1-Zn-N3 | 108.9(1) | 112.3(1) |
| X1-Zn-N13 | 104.6(1) | 108.0(1) |
| X2-Zn-N3 | 114.4(1) | 104.0(1) |
| X2-Zn-N13 | 107.2(1) | 110.2(1) |
| N3-Zn-N13 | 104.2(1) | 106.7(2) |
| N1-N2-N3 | 107.9(3) | 107.6(4) |

[a] X = Br, I.

**Table 2.** Selected interatomic distances (Å) and interatomic angles (°) for complexes tet-[Zn(NO_3)_2(Mebta)_2] (**5**) and oct-[Zn(NO_3)_2(Mebta)_2] (**6**).

| Complex 5 | | | |
|---|---|---|---|
| **Distances (Å)** | | **Angles (°)** | |
| Zn-O1 | 2.016(2) | O1-Zn-O4 | 87.9(1) |
| Zn···O2 | 2.588(5) | O1-Zn-N3 | 115.7(1) |
| Zn-O4 | 1.992(2) | O1-Zn-N13 | 108.3(1) |
| Zn···O6 | 2.722(6) | O4-Zn-N3 | 117.4(1) |
| Zn-N3 | 1.996(2) | O4-Zn-N13 | 105.0(1) |
| Zn-N13 | 1.995(2) | N3-Zn-N13 | 118.2(1) |
| N4-O1 | 1.290(3) | O2···Zn-O1 | 53.8(1) |
| N4-O2 | 1.228(3) | O2···Zn-O4 | 141.0(1) |
| N4-O3 | 1.214(3) | O2···Zn-N3 | 88.9(1) |
| N5-O4 | 1.291(3) | O2···Zn-N13 | 83.9(1) |
| N5-O5 | 1.214(3) | O2···Zn···O6 | 164.3(1) |
| N5-O6 | 1.229(3) | O6···Zn-O1 | 139.5(1) |
| N1-N2 | 1.336(3) | O6···Zn-O4 | 51.7(1) |
| N2-N3 | 1.319(3) | O6···Zn-N3 | 89.9(1) |
| N11-N12 | 1.332(3) | O6···Zn-N13 | 82.8(1) |
| N12-N13 | 1.313(3) | | |
| Complex 6 | | | |
| **Distances (Å)** | | **Angles (°)** [a] | |
| Zn-N3 | 2.040(2) | N3-Zn-O1 | 105.9(1) |
| Zn-O1 | 2.061(2) | N3-Zn-O1′ | 97.6(1) |
| Zn-O2 | 2.388(2) | N3-Zn-O2 | 92.5(1) |
| N1-N2 | 1.325(3) | N3-Zn-O2′ | 153.7(1) |
| N2-N3 | 1.323(3) | N3-Zn-N3′ | 101.7(1) |
| N4-O1 | 1.275(3) | O1-Zn-O1′ | 142.5(1) |
| N4-O2 | 1.244(3) | O1-Zn-O2 | 56.9(1) |
| N4-O3 | 1.220(3) | O1-Zn-O2′ | 93.9(1) |
| | | O2-Zn-O2′ | 84.0(1) |

[a] Primed atoms are generated by the symmetry operation $-x, y, -z + 1/2$.

**Table 3.** Selected bond distances (Å) and bond angles (°) for the cation of complex [Zn(Mebta)_4](ClO_4)_2 (**7**).

| **Bond Lengths (Å)** | | **Bond Angles (°)** | |
|---|---|---|---|
| Zn-N3 | 1.990(2) | N3-Zn-N13 | 111.0(1) |
| Zn-N13 | 1.987(2) | N3-Zn-N23 | 109.8(1) |
| Zn-N23 | 1.981(2) | N3-Zn-N33 | 108.6(1) |
| Zn-N33 | 2.007(2) | N13-Zn-N23 | 110.3(1) |
| N1-N2 | 1.325(3) | N13-Zn-N33 | 108.0(1) |
| N2-N3 | 1.320(3) | N23-Zn-N33 | 109.1(1) |

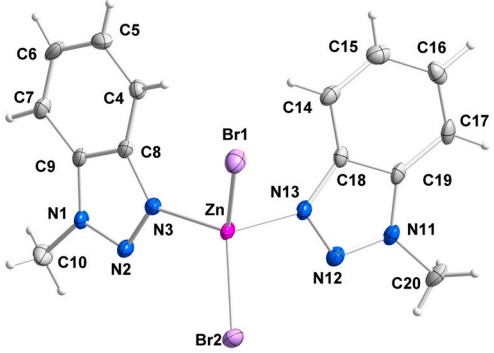

**Figure 2.** Labeled plot of the molecular structure of [ZnBr_2(Mebta)_2] (**3**). Thermal ellipsoids are presented at the 50% probability level.

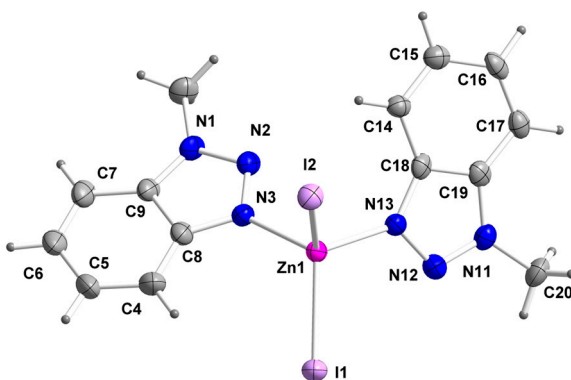

**Figure 3.** Labeled plot of the molecular structure of [ZnI$_2$(Mebta)$_2$] (**4**). Thermal ellipsoids are presented at the 50% probability level.

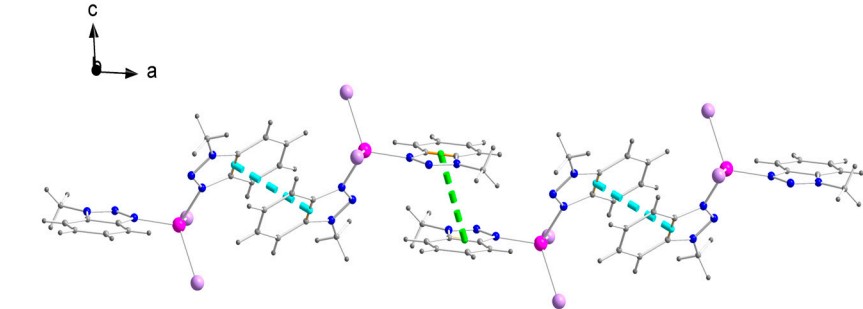

**(a)**

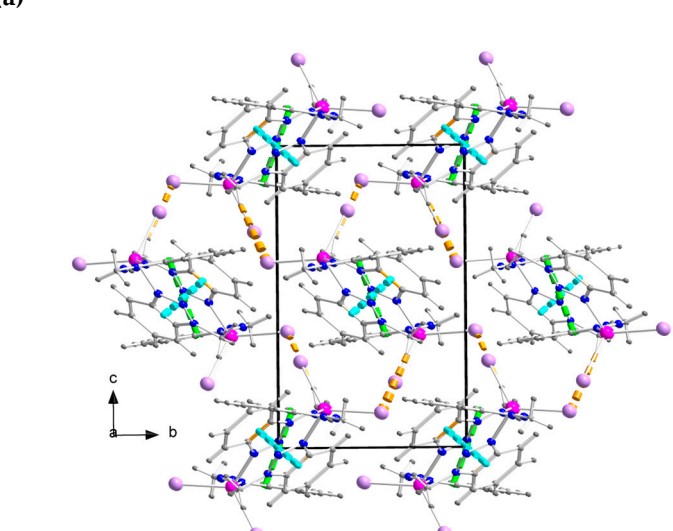

**(b)**

**Figure 4.** (**a**) Chains of [ZnI$_2$(Mebta)$_2$] molecules parallel to the *a* axis in the crystal structure of **4**. (**b**) Side view along the *a* axis. Light green and cyan dashed lines illustrate π–π stacking interactions. Orange dashed lines denote C10-H$_B$(C10)···I1 H bonds. See text for details.

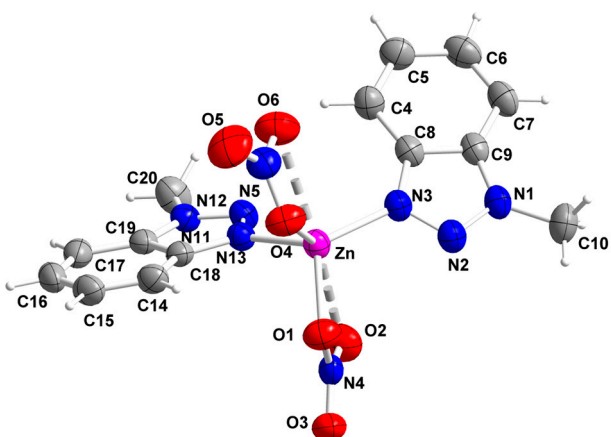

**Figure 5.** Labeled plot of the molecular structure of tet-[Zn(NO$_3$)$_2$(Mebta)$_2$] (**5**). The gray dashed lines denote very long Zn···O interactions. Thermal ellipsoids are presented at the 50% probability level.

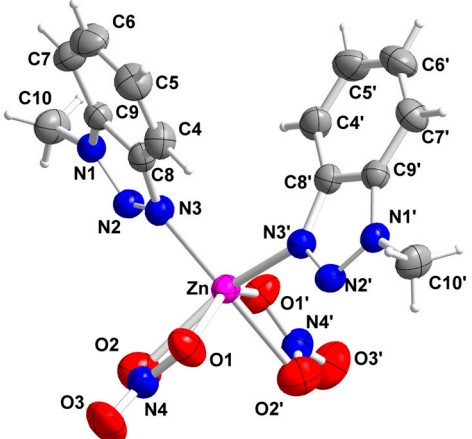

**Figure 6.** Labeled plot of the molecular structure of oct-[Zn(NO$_3$)$_2$(Mebta)$_2$] (**6**). Thermal ellipsoids are presented at the 50% probability level. Primed atoms are generated by the symmetry operation $-x, y, -z + 1/2$.

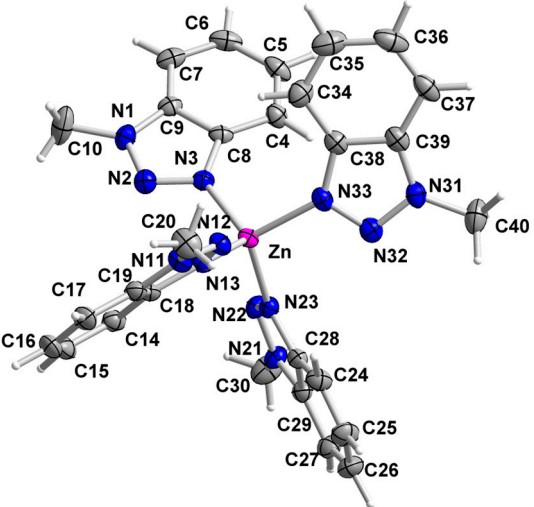

**Figure 7.** Labeled plot of the molecular structure of the cation that is present in complex [Zn(Mebta)$_4$](ClO$_4$)$_2$ (**7**). Thermal ellipsoids are presented at the 50% probability level.

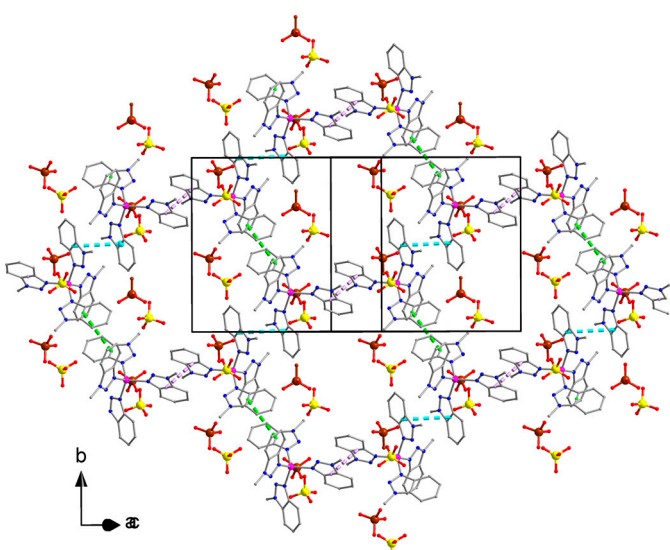

**Figure 8.** Layers of the [Zn(Mebta)$_4$]$^{2+}$ cations of **7** parallel to the (−101) crystallographic plane. Perchlorate anions are also shown, those with light colors lying above the layer and those with darker color lying below the layer. Violet, light green, and cyan dashed lines denote π–π interactions.

The molecular structures of **3** and **4** (Figures 2, 3, S1 and S2) are similar, but the complexes are not isomorphous. The crystals contain neutral [ZnX$_2$(Mebta)$_2$] (X = Br, I) molecules. The Zn$^{II}$ atom is coordinated by two halo (or halido) groups and two nitrogen atoms, the latter from two monodentate Mebta ligands. The molecules of Mebta utilize the N atoms at position 3 of the azole rings as donor sites. The ligand molecules are nearly planar. The coordination geometry of the metal ion is distorted tetrahedral, the donor atom–Zn$^{II}$-donor atom bond angles being in the ranges 104.2(1)–116.4(3)° and 104.0(1)–115.4(1)° for **3** and **4**, respectively. Interligand repulsions between the negatively charged and large halides presumably are responsible for the observed distortion; the Br-Zn-Br and I-Zn-I are the largest and have values of 116.4(3) and 115.4(1)°, respectively. The Zn$^{II}$–N bond lengths are typical for tetrahedral Zn(II) complexes [56]. The N-N bond distances are almost equal, ranging from 1.317(7) to 1.335(4) Å, implying that the —N═N—N< description of Mebta (Figure 1) might be slightly different due to coordination; this has as a consequence the small decrease of the double bond character between the nitrogen atoms at positions 2 and 3 of the azole ring. The decrease of the double bond character of the triazole ring would be expected as consequence of coordination. The Zn-I bond distances are larger than the Zn-Br ones by ca. 0.2 Å, as expected.

In the crystal structure of **3**, the [ZnBr$_2$(Mebta)$_2$] molecules are stacked along the *a* axis forming chains through π–π interactions involving the N1-, N2-, and N3-containing Mebta ligands. The distances of the planes of neighboring, centrosymmetrically related ligands are 3.54(1) Å (symmetry code: −*x* + 1, −*y*, −*z* + 1; light green dashed lines in Figure S3a) and 3.68(1) Å (symmetry code: −*x* + 2, −*y*, −*z* + 1; cyan dashed lines in Figure S3a). The other coordinated Mebta molecules (those containing N11, N12, N13, . . .), belonging to different chains, overlap through π–π interactions with their planes being 3.43(1) Å apart (symmetry code: −*x* + 2, −*y*, −*z* + 1; dark green dashed lines in Figure S3a), forming layers parallel to the (010) plane. Complex molecules belonging to neighboring chains also interact through C20(methyl carbon)-H(C20)···Br2 (*x* − 1, *y*, *z*) H bonds. The layers are stacked along the *b* axis and interact through C20···Br1 (*x* − 1, *y* + 1, *z*) H bonds, building the 3D architecture (Figure S3b).

In the crystal structure of **4**, neighboring [ZnI$_2$(Mebta)$_2$] molecules interact through alternating N1N2N3C8C4C5C6C7C9/N1N2N3C8C4C5C6C7C9 (cyan dashed lines) and N11N12N13C18C14C15C16C17C19/N11N12N13C18C14C15C16C17C19 (light green dashed lines) π–π overlapping and centrosymmetrically related pairs of Mebta forming chains

parallel to the *a* crystallographic axis (Figure 4a). The planes of the first pair of ligands are at 3.41(2) Å (symmetry code: $-x$, $-y-1$, $-z-1$) and those of the second are at 3.39(2) Å (symmetry code: $-x+1$, $-y+1$, $-z+1$). Chains of neighboring molecules interact through C10···I1 ($x$, $-y-3/2$, $z-1/2$) H bonds creating the 3D architecture of the structure (Figure 4b); the dimensions of this H bond are C10···I1 3.986(6) Å, $H_B$(C10)···I1 3.303(1) Å, and C10-$H_B$(C10)···I1 165.2(3)°.

Compound **5** (Figures 5 and S4) crystallizes in the triclinic space group $P\bar{1}$ with two tet-[Zn(NO$_3$)$_2$(Mebta)$_2$] molecules in the unit cell. The Zn$^{II}$ atom is coordinated by two monodentate Mebta ligands (again the donor atom is the nitrogen at position 3 of the azole ring) and two practically monodentate nitrato groups. The coordination geometry at the metal ion is very distorted tetrahedral, with the donor atom–Zn$^{II}$-donor atom angles being in the 87.9(1)–118.2(1)° range. The Zn···O2 and Zn···O6 interatomic distances are long, i.e., 2.588(5) and 2.722(6) Å, respectively. If these interactions are considered weakly bonding (in such a case, the nitrato groups are highly anisodentate), then the polyhedron around Zn$^{II}$ can be described as an extremely distorted octahedron; this viewpoint is illustrated in Figure 5. In the octahedral description of the coordination sphere, the *trans* pairs of the donor atoms cannot be defined and we thus favor the distorted tetrahedral description.

In the crystal structure of **5**, the tet-[Zn(NO$_3$)$_2$(Mebta)$_2$] molecules are stacked along the *c* axis forming chains through π–π interactions with the participation of the N1N2N3C8C4C5C6C7C9 coordinated Mebta molecules. The distances between the planes of neighboring, centrosymmetrically related ligands are 3.574(2) Å (symmetry code: $-x+1$, $-y$, $-z$; light green dashed lines in Figure S5a) and 3.496(3) Å (symmetry code: $-x+1$, $-y$, $-z+1$; cyan dashed lines in Figure S5a). The Mebta ligands defined by atoms N11N12N13C18C14C15C16C17C19 of complex molecules belonging to neighboring chains overlap through π–π interactions, with the distance between the planes being 3.340(6) Å (symmetry code: $-x+1$, $-y+1$, $-z$; dark green dashed lines in Figure S5a), forming layers parallel to the (100) plane (Figure S5a). The layers are stacked along the *a* axis and interact through C20···O3 ($x+1$, $y$, $z$) H bonds, building the 3D structure (Figure S5b).

Complex **6** crystallizes in the monoclinic space group $C2/c$ with four molecules in the unit cell. Its structure consists of well separated oct-[Zn(NO$_3$)$_2$(Mebta)$_2$] (Figures 6 and S6) molecules with two chelating nitrato groups and two monodentate Mebta ligands. The Zn$^{II}$ center sits on a crystallographic two-fold axis of symmetry ($C_2$) which bisects the N3-Zn-N3′ angle. The coordination geometry of the metal ion is distorted octahedral. The Zn$^{II}$-O$_{nitrato}$ bonds *trans* to N3 [2.388(2) Å] are longer than those *trans* to O [2.061(2) Å], thus resulting in a *cis-cis-trans* configuration. The difference in Zn-O bond lengths is due to the fact that in *cis-cis-trans*-bis(bidentate ligand)bis(monodentate ligand) metal complexes, one end donor site of the bidentate ligands is associated with less repulsion energy than the other [50]. The significant distortion of the octahedron is a consequence of the bidentate chelating nitrato groups and their small bite angle [56.9(1)°]. The nitrato ligand is planar, the sum of the O-N-O bond angles being exactly 360°. As expected, the O1-N4-O2 angle involving the coordinated oxygen atom is the smallest [116.1(2)°]. The chelating nature of the nitrato groups implies that the N-O$_{not-coordinated}$ bond length is shorter [1.220(3) Å] than the N-O$_{coordinated}$ bond distances [1.244(3) and 1.275(3) Å]. The Zn$^{II}$-N bond lengths in **6** [2.040(2) Å] are somewhat longer than the corresponding ones in **5** [1.995(2) and 1.996(2) Å], due to the higher coordination number of the metal ion in the former (six) than in the latter (four).

Complex **6** is isomorphous with compound oct-[Co(NO$_3$)$_2$(Mebta)$_2$] [50]. The only difference is that the Co(II) derivative exhibits a more symmetric coordination of the nitrato ligand than **6** [Co-O 2.059(2) and 2.277(2) Å; Zn-O 2.061(2) and 2.388(2) Å]. A simple indication of the increase in the bidentate character of the nitrato groups according to the series Zn$^{II}$ < Co$^{II}$ is provided by observing the decrease in the M-O1-N bond angle; this bond angle is 101.8(2)° in **6** and 97.8(2)° for the Co(II) complex.

In the lattice structure of **6**, the presence of π–π stacking interactions between Mebta ligands belonging to neighboring complex molecules result in the formation of zig-zag chains running parallel to the *c* axis (Figure S7a). The centrosymmetrically related ligands display an interplane distance of 3.545(4) Å (symmetry code: $-x$, $-y$, $-z + 1$; light green dashed lines in Figure S7a). The metal–metal distance within the chain is 9.087(2) Å. These zig-zag chains are further associated through C4···O3 ($-x - 1/2$, $y - 1/2$, $-z + 1/2$) H bonds (orange dashed lines in Figure S7b), creating the overall 3D framework of the structure.

The crystal structure of **7** consists of $[Zn(Mebta)_4]^{2+}$ and $ClO_4-$ ions in a 1:2 ratio. Selected bond lengths (Å) and angles (°) are given in Table 3. In the asymmetric unit of the cell of the compound, which crystallizes in the monoclinic space group *I2/a*, there is a mononuclear $[Zn(Mebta)_4]^{2+}$ cation, one $ClO_4-$ at general position, and two other $ClO_4^{-}$ ions at positions where the chlorine atoms occupy sites with 2-fold symmetry, resulting in two anions per formula unit and thus ensuring charge neutrality. The perchlorate anions are tetrahedral. The $Zn^{II}$ center is coordinated to four Mebta ligands in a tetrahedral fashion. The tetrahedral coordination is almost ideal; the Zn-N bond lengths and N-Zn-N bond angles are in the ranges 1.981(2)–2.007(2) Å and 108.0(1)–111.0(1)°, respectively.

In the lattice of **7**, the presence of π–π stacking interactions between Mebta ligands of neighboring cations results in the formation of layers parallel to the ($-101$) crystallographic plane with a brick wall arrangement (Figure 8). Three of the Mebta ligands on each cation interact with the corresponding ligands of three neighboring cations through centers of symmetry. The interplane distance between the N1N2N3C8C4C5C6C7C9/N1N2N3C8C4C5C6C7C9 ($-x + 3/2$, $-y + 1/2$, $-z + 1/2$) ligands is 3.318(8) Å (violet dashed lines in Figure 8); that between the N21N22N23C28C24C25C26C27C9 and its symmetry-related ($-x + 1$, $-y + 1$, $-z$) ligands is 3.37(1) Å (light green dashed lines in Figure 8); and that between the N31N32N33C38C34C35C36C37C39 and its symmetry-related partner ($-x + 1$, $-y$, $-z$) is 3.368(3) Å (cyan dashed lines in Figure 8). $ClO_4^{-}$ ions are located above and below the layers, which either participate in the formation of the layers or serve as bridging units of neighboring layers, both types through C-H···$O_{perchlorate}$ H bonds.

Crystals of $[Zn(Mebta)_4](PF_6)_2$ (**8**) were also obtained but they showed poor diffraction. Repeating efforts to improve the quality of the crystals proved unsuccessful. However, we were able to collect data in order to establish the gross structural features of the complex. Some crystallographic data: Empirical formula $C_{28}H_{28}ZnP_2F_{12}N_{12}$; formula weight 887.93; monoclinic space group *I2/a*; *a* = 17.6543(3) Å, *b* = 15.3889(2) Å, *c* = 25.9892(5) Å; β = 96.214(1)°; *V* = 7019.3(2) Å$^3$; *Z* = 8. The data revealed that the structure of the complex consists of mononuclear tetrahedral $[Zn(Mebta)_4]^{2+}$ cations and $PF_6^{-}$ anions. The structure of the cation is almost identical with that of complex **7**. There are three crystallographically independent $PF_6^{-}$ in the unit cell. One sits in a general position and the other two possess a 2-fold axis of symmetry. The former contributes one and the latter two each contribute half $PF_6^{-}$ ions in the asymmetric unit of the cell, resulting in a total number of two anions. The fluorine atoms of the $PF_6^{-}$ ion at the general position are disordered over two sites.

## 2.3. Spectroscopic and Physical Characterization in Brief

The complexes were characterized by a variety of physical and spectroscopic techniques in solution and in the solid state. Detailed assignments of the various spectroscopic features are provided in Section 3.2 under the preparation of each complex. Selected vibrational data (cm$^{-1}$) are also listed in Tables 4 and 5 for convenience. A few spectra are shown in Figures 9, 10 and S9–S16. The assignments are based on extensive literature studies, comparison of the spectroscopic characteristics between free Mebta and the complexes, and between complexes of a given family (e.g., **1–3**), and our previous experience with coordination complexes of Mebta [49–53]. Here, we discuss some data of diagnostic value.

**Table 4.** Selected diagnostic vibrational spectral data for Mebta and complexes **1-8** [a].

| Compound | $\nu_{as}$(NNN) | $\nu_s$(NNN) | $\nu_{as}$(ZnX)$_t$ | $\nu_s$(ZnX)$_t$ | $\nu$(ZnO)$_t$ |
|---|---|---|---|---|---|
| Mebta | 1197s (1180w) | 1110m (1105s) | | | |
| 1 | 1218s (1226m) | 1136m (1140s) | 324s (325m) | 300s (300s) | |
| 2 | 1222s [b] | 1138m [b] | | | |
| 3 | 1218s (1224m) | 1135m (1140s) | 243s [c] (246m [c]) | 219s (220s) | |
| 4 | 1219s (1233w, 1224m) | 1132m (1136s) | 208s (208m) | 189s (182s) | |
| 5 | 1220s [b] | 1139m [b] | 348s [b] | 300m [b] | |
| 6 | 1220s (1228m) | 1137m (1141m) | | | 326w, 292sh, 274s [d], 255sh (326m, 270s [d], 260m) |
| 7 | 1218m [b] | 1144s [b] | | | |
| 8 | 1226m [b] | 1170w [b] | | | |

[a] The corresponding Raman peaks are written within parentheses. [b] No Raman spectra available. [c] Most probably overlapping vibration with $\nu_s$(ZnN) [Table 5]. [d] Most probably overlapping vibration with $\nu$(ZnN) [Table 5]. Abbreviations: s = strong, m = medium, w = weak, sh = shoulder, X = Cl, Br, I, ONO$_2$ (monodentate).

**Table 5.** Selected diagnostic vibrational spectral data for complexes **1–8** concerning mainly zinc(II)-nitrogen stretching vibrations [a].

| Compound | $\nu$(ZnCl)$_t$($T_2$)/ $\nu$(ZnN)($T_2$) | $\nu_{as}$(ZnN) | $\nu_s$(ZnN) | $\nu$(ZnN) |
|---|---|---|---|---|
| Mebta | | | | |
| **1** | | 274m (270m) | 247m (249s) | |
| **2** | 293s [b] | | | |
| **3** | | 275s (271m) | 243s [c] (246m [c]) | |
| **4** | | 263m (263m) | 248m (246m) | |
| **5** | | 291s [b] | 248m [b] | |
| **6** | | | 263m [b] | 274s [d], 242m (270s [d], 248m) |
| **7** | 288s [b] | | | |
| **8** | 287s [b] | | | |

[a] The corresponding Raman peaks are written within parentheses. [b] No Raman spectra available. [c] Most probably overlapping vibration with $\nu_{as}$(ZnX)$_t$ [Table 4]. [d] Most probably overlapping vibration with $\nu$(ZnO)$_t$ [Table 4]. Abbreviations: s = strong, m = medium, w = weak, sh = shoulder.

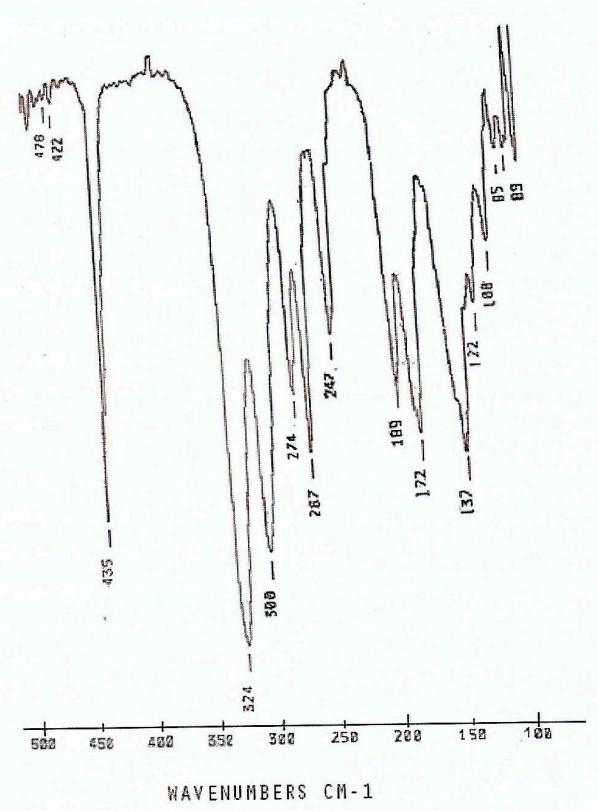

**Figure 9.** The FT far-IR spectrum (polyethylene, cm$^{-1}$) of [ZnCl$_2$(Mebta)$_2$] (**1**).

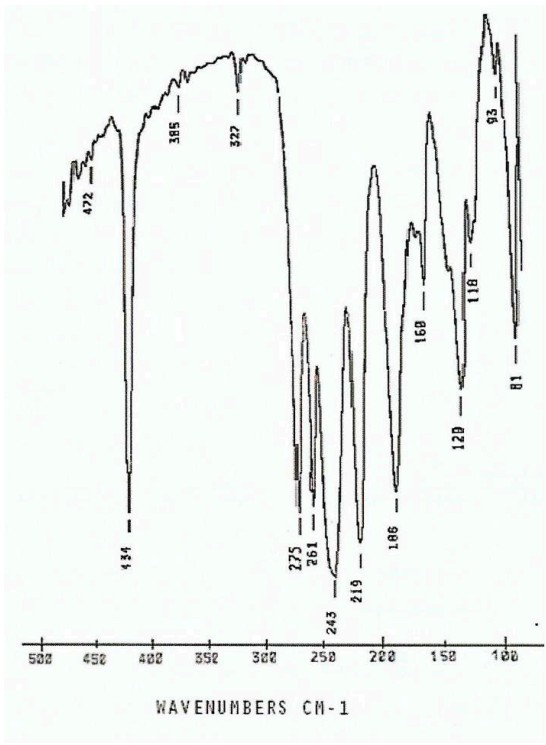

**Figure 10.** The FT far-IR spectrum (polyethylene, cm$^{-1}$) of [ZnBr$_2$(Mebta)$_2$] (**3**).

The molar conductivity values ($\Lambda_M$) of **2**, **7**, and **8** are 153 (in MeNO$_2$), 71 (in DMSO) and 79 (in DMSO) S cm$^2$ mol$^{-1}$ suggesting 1:2 electrolytes in these solvents [57], as expected from their ionic formulae. Somewhat to our surprise, the $\Lambda_M$ values for **1**, **3**, **4**, and **5** in DMSO (68–81 S cm$^2$ mol$^{-1}$) are also indicative of 1:2 electrolytes. This suggests decomposition of the complexes, most probably through the reaction represented by Equation (9), where X = Cl, Br, I, NO$_3$, and x = 4–6. This decomposition scheme is also supported by NMR spectroscopy (vide infra). In accordance with our proposal, the $\Lambda_M$ values of **1**, **3**, **4**, and **6** in MeNO$_2$ (this solvent has a negligible capacity for coordination) are 10, 12, 14, and 15 S cm$^2$ mol$^{-1}$, suggesting essentially non-electrolytes [57]. It should be mentioned at this point that the $\Lambda_M$ values of some neutral complexes (**1**, **3–5**) in DMSO appear to be intermediate between those reported [57] for 1:1 and 1:2 electrolytes. This might indicate the presence of singly charged species in solution, e.g., [Zn(DMSO)$_{x-1}$X]$^+$ and X$^-$. Thus, the decomposition scheme illustrated in Equation (9) can be considered as oversimplified.

$$[ZnX_2(Mebta)_2] + x\,dmso \xrightarrow{d_6-DMSO} [Zn(d_6-DMSO)_x]^{2+} + 2X^- + 2Mebta \qquad (9)$$

The $^1$H NMR spectrum of Mebta in $d_6$-DMSO shows a singlet signal (3H) at δ 4.34 ppm, a triplet signal (1H) at δ 7.42 ppm, a triplet (1H) at δ 7.57 ppm, a doublet (1H) at δ 7.86 ppm, and a doublet (1H) at δ 8.06 ppm, assigned to the protons attached to carbons at the positions 10 (-CH$_3$), 5, 6, 7, and 4 (see numbering scheme in Figure 1), respectively [49,58–60]. The $^{13}$C[$^1$H] NMR spectrum of Mebta in $d_6$-DMSO shows resonances (assignments in parentheses) at δ 34.2(CH$_3$), 119.1(C4), 124.0(C5), 110.7(C7), 145.3(C8), 133.5(C9), and 127.2(C6) [11,49,60]; assignments were also aided by the APT (Attached Proton Test) technique. The $^1$H and $^{13}$C[$^1$H] NMR spectra of **1**, **3**, **4**, **5**, **6**, **7**, and **8** in $d_6$-DMSO are identical to those of free Mebta, supporting their decomposition in solution. The decomposition of **7** and **8**, which was not reported earlier (due to their ionic character which makes the $\Lambda_M$ values not diagnostic), can be represented by Equation (10), X = ClO$_4$, PF$_6$, and x = 4–6.

$$[Zn(Mebta)_4]X_2 + x(d_6-DMSO) \xrightarrow{d_6-DMSO} [Zn(d_6-DMSO)_X]^{2+} + 2X^- + 2Mebta \quad (10)$$

Thermal decomposition data (TG/DTG) are available for compounds **1–4** and **6**. For (MebtaH)$_2$[ZnCl$_4$] (**2**), the data show a mass loss of 16.0% between 150 and 205 °C corresponding to the elimination of two moles of HCl per mole of the complex (theoretical value: 15.3%). The IR and far-IR spectra of the solid obtained after heating at 210 °C are identical with those of [ZnCl$_2$(Mebta)$_2$] (**1**), supporting the proposed decomposition illustrated by Equation (3). At temperatures above 220 °C, the solid sample decomposes in the same manner as the authentic complex **1**. The solid complexes **1** and **3** decompose in a similar manner. The first decomposition occurs at ~170–230 °C and corresponds to the loss of one mole of Mebta per mole of complex (experimental values: 33.0% for **1** and 27.2% for **3**; theoretical values: 33.1% for **1** and 27.1% for **3**). There is a small plateau between ca. 230 and 250 °C, and at higher temperatures, the decomposition takes place in two steps up to 650 °C. The thermally stable ZnX$_2$(Mebta) (X = Cl, Br) intermediates **1a** and **3a**, obtained after an isothermal TG experiment at 230 °C, were isolated and studied by IR and far-IR spectra (vide infra). Attempts to recrystallize and obtain single crystals of **1a** and **3a** were unsuccessful. The iodido complex **4** decomposes similarly, but there is no plateau after the first Mebta elimination, indicating that a species with the empirical formula ZnI$_2$(Mebta) is not thermally stable. Complex oct-[Zn(NO$_3$)$_2$(Mebta)$_2$] is thermally stable up to 175 °C. The decomposition occurs in two steps (with no indication of one or two Mebta elimination) and the final plateau > 250 °C probably corresponds to ZnO as evidenced by mass calculations (experimental value at 650 °C: 18.0%; theoretical value for ZnO as a final residue: 17.9%). We did not further pursue the characterization of the final residue.

The presence of H-bonded MebtaH$^+$ ions in (MebtaH)$_2$[ZnCl$_4$] (**2**) is proven by the appearance of the characteristics broad ν(N$^+$H) IR band at 3200–2830 cm$^{-1}$ [50]. The IR spectrum of free Mebta exhibits two strong-to-medium intensity bands at 1197 and

1110 cm$^{-1}$ associated with the $\nu_{as}$(NNN) and $\nu_s$(NNN) modes, respectively [52]. These bands/peaks are shifted to ca. 1220 and 1140 cm$^{-1}$, respectively, in the spectra of **1** and **3-7**; the same trend is observed in the spectrum of **2** that contains the 1-methylbenzotriazolium cation. These shifts are compatible, with N3 (see the numbering scheme of Figure 1) being the donor atom to Zn$^{II}$ or the protonation site [50,52]. The monodentate coordination of Mebta is also responsible for the two IR bands at ca. 780 and 750 cm$^{-1}$; the lower wavenumber band is stronger and assigned to a C-H out-of-plane vibration, and the higher wavenumber one is due to a vibration involving both triazole and benzene in-plane bending [52]. The corresponding Raman peaks are located at ca. 780 and 745 cm$^{-1}$.

The far-IR spectra of **1** and **3–5** (Figures 9 and 10 for **1** and **3**, respectively) exhibit the two $\nu$(ZnX) (X = Cl, Br, I, ONO$_2$) and the two $\nu$(ZnN) bands in the regions expected for pseudotetrahedral C$_{2v}$ point group symmetry [61]; the corresponding Raman peaks are visible at approximately the same wavenumbers. In the spectrum of **2**, a very intense band is assigned to the asymmetric stretching T$_2$ mode of [ZnCl$_4$]$^{2-}$ (assuming T$_d$ point group symmetry). The same symmetry also applies to **7** and **8** and the strong band at ca. 290 cm$^{-1}$ is attributed to the asymmetric T$_2$ mode of [Zn(Mebta)$_4$]$^{2+}$ [61]. The higher wavenumbers of $\nu$(ZnN) in **5** compared with those in **6** suggest stronger Zn-N bonds in the former than in the latter (confirmed by crystallography), a consequence of the lower coordination number of Zn$^{II}$ in **5** than in **6** (four vs. six). The molecule of **6** can be approximately considered as cis (C$_{2v}$) of the MX$_4$Y$_2$ type. This is expected to give four Zn-O stretching (2A + B$_1$ + B$_2$) and two Zn-N stretching (A$_1$ + B$_1$) modes, all of which are IR- as well as Raman- active. These modes are actually observed (Table 4), the only difference being the appearance of three (instead of four) Zn-O stretches.

The spectrum of **7** (Figure S13) exhibits strong bands at 1112 and 630 cm$^{-1}$ attributable [61] to the $\nu_3$(F$_2$)[$\nu_d$(ClO)] and $\nu_4$(F$_2$)[$\delta_d$(OClO)] vibrations of the uncoordinated tetrahedral T$_d$ ClO$_4^-$ ions. The PF$_6^-$ ion belongs to O$_h$ point group and should exhibit the IR-active vibrations $\nu_3$(F$_{1u}$)[$\nu$(PF)] and $\nu_4$(F$_{1u}$)[$\delta$(FPF)]; these are located [61] at 834 and 558 cm$^{-1}$, respectively, in the IR spectrum of **8** (Figure S14).

Finally, the IR spectra of **5** and **6** can differentiate between the monodentate (**5**) and bidentate (**6**) nitrato groups. In the former, the bands at 1450 and 1309 cm$^{-1}$ are assigned to the $\nu_5$(B$_2$)[$\nu_{as}$(NO$_2$)] and $\nu_1$(A$_1$)[$\nu_S$(NO$_2$)], respectively; their rather small separation (141 cm$^{-1}$) is indicative of monodentate nitrates [61]. By contrast with **5**, the IR spectrum of **6** shows the characteristic bands of bidentate nitrato ligands. The bands at 1500 (overlapped with an aromatic stretch) and 1275 cm$^{-1}$ are assigned to the $\nu_1$(A$_1$)[$\nu$(N=O)] and $\nu_5$(B$_2$)[$\nu_{as}$(NO$_2$)] modes, with their large separation (225 cm$^{-1}$) being indicative of bidentate nitrates [61]. In some cases, the IR spectra of the two nitrato complexes exhibit a variable intensity band at 1384 cm$^{-1}$, characteristic of the $\nu_3$(E')[$\nu_d$(NO)] mode of the planar ionic nitrate [61]; such nitrates do not exist in the crystal structures of **5** and **6**. The appearance of this band indicates that an amount of the nitrato ligands in the samples are replaced by bromides that are present in excess in the KBr matrix (used for the preparation of the pellets), thus producing ionic nitrates (KNO$_3$) [62,63]; this replacement is facilitated by the pressure used. In accordance with this conclusion, the band at 1384 cm$^{-1}$ is always absent from the Raman spectra of **6** and it appears more often in the IR spectrum of **5** in which the monodentate nitrato ligands are replaced more easily by bromides compared with **6**, which contains bidentate nitrates.

The far-IR spectra of solids **1a** and **3a** with the empirical formula ZnX$_2$(Mebta) (X = Cl, **1a**; X = Br, **3a**), obtained from the thermal decomposition of **1** and **3**, respectively, exhibit far-IR bands attributable to both terminal and bridging halido (Cl$^-$, Br$^-$) ligands. Thus, the spectrum of **1a** exhibits strong bands at 350 and 229 cm$^{-1}$ which are due to the $\nu$(ZnCl)$_t$ and $\nu$(ZnCl)$_b$ vibrations [61], respectively. Similarly, the strong bands at 240 and 167 cm$^{-1}$ in the spectrum of **3b** are assigned to the $\nu$(ZnBr)$_t$ and $\nu$(ZnBr)$_b$ modes [61], respectively. The $\nu$(ZnN) vibrations appear at 243 (**1a**) and 263 (**3a**) cm$^{-1}$. These spectroscopic features suggest polymeric structures for **1a** and **3a** possessing both terminal and bridging halido

groups; we tentatively propose a $\{Zn^{II}(X_t)(X_b)_2N\}$ coordination sphere with tetrahedral coordination at $Zn^{II}$.

### 2.4. Theoretical Calculations

Prompted by the experimental facts that (i) Mebta always behaves as a monodentate ligand with N3 as the donor atom [50–53] and present work; and (ii) the nitrato complexes **5** and **6**, although isostoichiometric, exhibit two different stereochemistries, we performed a quantum-chemical investigation in an attempt to understand these aspects. Thus, a theoretical investigation was undertaken at the ab initio level (MP2) in ethanolic solution of Mebta (EtOH is the most often used solvent for the preparation of its metal complexes), followed by DFT studies in the gas phase for the two nitrato complexes. Data are presented in Figures 11, S17 and S18, and Tables 6 and S1–S6.

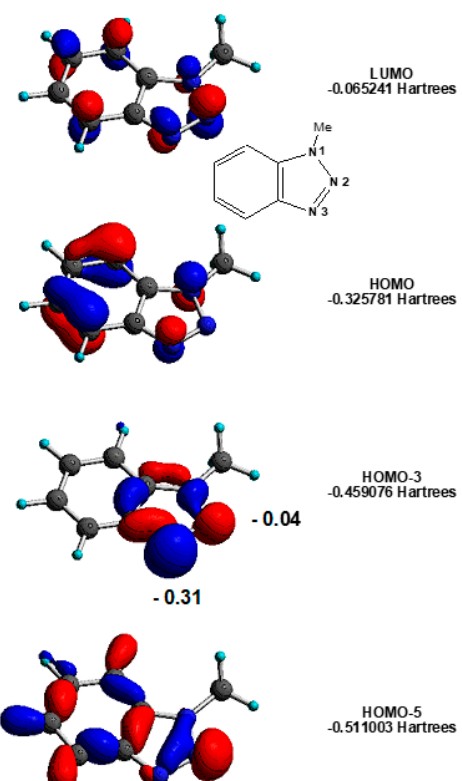

**Figure 11.** MP2/6-31+G* calculated FMOs of Mebta in EtOH.

Concerning the former issue, the most salient feature of the ground-state electronic structure of Mebta is its three (out of five) highest energy filled orbitals OMOs HOMO, HOMO-3, and HOMO-5, which are schematically depicted in Figure 11. Unlike HOMO which is a π-type orbital, both HOMO-3 and HOMO-5 are σ-type mostly "lone pair" on N3 and N2, respectively. Moreover, contrary to HOMO-3, centered mostly on N3 and being only about 0.1 a.u. lower in energy than the HOMO, HOMO-5 (centered mostly on N2) is about 0.2 a.u. lower. The latter energy differences, along with (a) the σ-type nature of the two HOMOs (b) the high frontier density $P_r$(HOMO-3) [64] and $P_r$(HOMO-5) values, calculated for the N3 and N2 atoms (0.41 and 0.29, respectively), and (c) the high calculated, negative neutral atomic charge [65–67] for N3 only ($q_r = -0.31$ |e|) (that of N2 is $-0.04$ |e|), strongly suggest that only the N3 atom should be a nucleophilic center for $ML_{m-1}^{n+}$ attack of the Mebta ligand. In particular, this nucleophilic attack, possibly being a HOMO-3 (Mebta)-LUMO ($ML_{m-1}^{n+}$) interaction, should be mostly charge- rather than frontier orbital-controlled.

**Table 6.** Mulliken population analysis (BSII) atom by atom data and Wiberg bond indices (BSII) for the coordination bonds [a] of the nitrato complexes **5** and **6**.

| Complex | Zn-Donor Atom | Bond Length (Å) | Mulliken Pop Anal | Wiberg Bond Indices |
|---|---|---|---|---|
| 5 | Zn-N3 | 2.04 | 0.2233 | 0.2705 |
| | Zn-N13 | 2.06 | 0.1871 | 0.2588 |
| | Zn-O1 | 1.99 | 0.2065 | 0.2794 |
| | Zn-O4 | 2.00 | 0.1770 | 0.2663 |
| | Zn-O2 | 2.51 | 0.0723 | 0.1021 |
| | Zn-O6 | 2.66 | 0.0504 | 0.0723 |
| 6 | Zn-N3 | 2.15 | 0.1568 | 0.2261 |
| | Zn-N3′ | 2.15 | 0.1568 | 0.2261 |
| | Zn-O1 | 2.08 | 0.1535 | 0.2250 |
| | Zn-O1′ | 2.08 | 0.1535 | 0.2250 |
| | Zn-O2 | 2.19 | 0.1336 | 0.1911 |
| | Zn-O2′ | 2.19 | 0.1336 | 0.1911 |

[a] The bond lengths are the theoretical values; the atom numbering scheme corresponds to the experimental one (Figures 5 and 6).

Although comparison between gas-phase calculations and single-crystal crystallographic studies is not allowed, we see that the DFT gas-phase structures of both **5** and **6** are almost identical with the experimental ones (Figures S17 and S18). Moreover, both BSI and BSII models used afforded almost identical structural data (Tables S1 and S2), closely resembling the crystallographic data. Additionally, based upon the Mulliken population analysis atom by atom data and the Wiberg bond indices, conclusions for the coordination bond strengths could also be drawn. Based on the data shown in Table 6, it is seen that the Zn-N bonds appear to be of rather comparable strength with the shorter Zn-O ones, in close agreement with results from several Zn(II) complexes containing nitrato ligands and monodentate N-donors. It should be also stressed that in both complexes, the longer Zn-O bonds or bonding interactions appear as weak ones (since they present small positive corresponding values) in close agreement with the experimental structural data. Finally, based upon the sum of electronic and thermal enthalpy values, complex **6** appears thermodynamically slightly more stable by ca. 0.7 kcal/mol (BSII) than **5** in the gas phase (Table S6).

## 3. Materials and Methods

### 3.1. Materials and Instrumentation

All manipulations were performed under aerobic conditions. Reagent and solvents were purchased from Alfa Aesar and Sigma-Aldrich and used as received. The purity of 1-methylbenzotriazole (Mebta) was checked by $^1$H NMR spectroscopy. Deionized water was received from in-house facility. **Safety note:** $Zn(ClO_4)_2 \cdot 6H_2O$, Mebta, and its complexes are potentially explosive, and caution should be exercised in handling such materials. The use of plastic spatula is recommended. The small quantities used in this study did not cause any problem.

Carbon, hydrogen, and nitrogen microanalyses were performed by the Instrumental Analysis Center of the University of Patras. Conductivity measurements were carried out at 25 °C with a Metrohm-Herisau E-527 bridge and a cell of standard constant. Thermogravimetric (TG) and differential thermogravimetric (DTG) data (30–650 °C) were obtained on a Dupont R90 device equipped with a 951 TG analyzer under a $N_2$ gas flow (50 cm$^3$ min$^{-1}$). Sample weights of 10–15 mg and heating rates of 1 and 5 °C min$^{-1}$ were used. FT IR spectra (4000–450 cm$^{-1}$) were recorded using a PerkinElmer 16PC spectrometer with samples prepared as KBr pellets under pressure. FT far-IR spectra (400–40 cm$^{-1}$) were recorded as a Bruker IFS 113v spectrometer using polyethylene pellets. Raman spectra were obtained using a Spex spectrometer (model 1403) equipped with an Ar$^+$ laser of Spectra Physics (model 2020). The excitation was performed at 514.5 nm with an output power of 60–300 mW de-

pending on the sample. Some spectra were recorded with an Instruments S.A. spectrometer equipped with a $Kr^+$ laser of Spectra Physics (model 164); the excitation was performed at 647.1 nm. $^1H$ NMR spectra in $d_6$-DMSO (isotopic purity >99.5%) were recorded on Varian EM-390 (90 MHz), FT Bruker WM-250 (250 MHz), and FT Bruker AM-400 (400.13 MHz) spectrometers; the signal of TMS was used as reference. Proton-decoupled $^{13}C$ spectra in $d_6$-DMSO were recorded on a Varian XL-300 spectrometer using 1,4-dioxane as reference compound. Some $^{13}C[^1H]$ NMR spectra were also recorded applying the APT technique.

### 3.2. Syntheses of the Complexes

#### 3.2.1. [ZnCl$_2$(Mebta)$_2$] (1)

Method (a): To a stirred, colorless solution of $ZnCl_2$ (0.034 g, 0.25 mmol) in EtOH (10 mL) was added solid Mebta (0.067 g, 0.50 mmol). The solid soon dissolved and the resulting solution was stirred for a further 10 min during which time a very small quantity of a beige solid appeared. The reaction mixture was filtered and the filtrate stored at $-10\ °C$. After 1 d, colorless crystals of the product were precipitated, which were collected by filtration, washed with cold EtOH (1 mL) and $Et_2O$ (2 × 2 mL), and dried in air. Yield: 35%. Anal. Calcd. (%) for $C_{14}H_{14}N_6ZnCl_2$: C, 41.76; H, 3.51; N, 20.88. Found (%): C, 41.52; H, 3.57; N, 21.18. $\Lambda_M$ (DMSO, 25 °C, $10^{-3}$ M) = 70 S cm$^2$ mol$^{-1}$. $\Lambda_M$ (MeNO$_2$, 25 °C, $10^{-3}$ M) = 10 S cm$^2$ mol$^{-1}$. Selected IR and far-IR data (cm$^{-1}$): 1218s, $\nu_{as}$(NNN); 1136m, $\nu_s$(NNN); 324s, $\nu_{as}$(ZnCl)$_t$; 300s, $\nu_s$(ZnCl)$_t$; 274m, $\nu_{as}$(ZnN); 240m, $\nu_s$(ZnN). Selected Raman peaks (cm$^{-1}$): 1226m, $\nu_{as}$(NNN); 1140s, $\nu_s$(NNN); 325m, $\nu_{as}$(ZnCl)$_t$; 300s, $\nu_s$(ZnCl)$_t$; 270m, $\nu_{as}$(ZnN); 249s, $\nu_s$(ZnN). $^1H$ NMR ($d_6$-DMSO, δ/ppm; since the spectrum of the complex is identical with that of free Mebta, the numbering scheme of protons follows Figure 1): 8.06 (d, 2H/H4), 7.86 (d, 2H/H7), 7.57 (t, 2H/H6), 7.42 (t, 2H/H5), 4.34 (s, 6H/-CH$_3$). $^{13}C[^1H]$ NMR ($d_6$-DMSO, δ/ppm; since the spectrum of the complex is identical with that of free Mebta, the numbering scheme of carbons is the same with that used in Figure 1): 145.21(C8), 133.45(C9), 127.25(C6), 124.03(C5), 119.09(C4), 110.71(C7), 34.30(C10).

Method (b): Complex (MebtaH)$_2$[ZnCl$_4$] (2) decomposes at 150–205 °C under N$_2$. The mass loss corresponds to the elimination of two moles of HCl per mole of **2** (experimental value: 16.0%; theoretical value: 15.3%). A plateau is visible at 205–220 °C. The thermally stable intermediate, obtained after an isothermal TG experiment at 210 °C, was collected. The IR, far-IR, and $^1H$ NMR spectra of this intermediate are identical with those of authentic **1** isolated with method (a).

#### 3.2.2. (MebtaH)$_2$[ZnCl$_4$] (2)

$ZnCl_2$ (0.260 g, 1.91 mmol) and Mebta (0.510 g, 3.80 mmol) were easily dissolved in concentrated hydrochloric acid (5 mL, 60 mmol). The resulting pale-yellow solution was allowed to slowly evaporate at ~50 °C (water bath) in the hood until it reached a volume of 3 mL. The solution was cooled in an ice bath for a few hours. A solid was precipitated which was collected by filtration, washed with cold EtOH (2 × 3 mL) and $Et_2O$ (5 × 5 mL), and dried in air overnight. The solid decomposes slowly in the normal laboratory atmosphere releasing gaseous HCl. Yield: 42%. Anal. Calcd. (%) for $C_{14}H_{16}N_6ZnCl_4$: C, 35.36; H, 3.40; N, 17.68. Found (%): C, 35.78; H, 3.33; N, 17.19. $\Lambda_M$ (MeNO$_2$, 25 °C, $10^{-3}$ M) = 153 S cm$^2$ mol$^{-1}$. Selected IR and far-IR data (cm$^{-1}$): 3202mb, 3097mb, and 3059mb, $\nu$(N$^+$H); 1222s, $\nu_{as}$(NNN); 1138m, $\nu_s$(NNN); 293s, $\nu$(ZnCl)$_t$ (T$_2$ in T$_d$). The ionic complex is insoluble in DMSO and no satisfactory $^1H$ NMR spectra could be obtained.

#### 3.2.3. [ZnBr$_2$(Mebta)$_2$] (3)

To a solution of $ZnBr_2$ (0.056 g, 0.25 mmol) in EtOH (4 mL) was added a solution of Mebta (0.067 g, 0.50 mmol) in the same solvent (4 mL). The resulting solution was stirred for 30 min, during which time a very small amount of a beige solid appeared. The reaction mixture was filtered and the filtrate was left undisturbed in a closed flask. X-ray quality colorless crystals of the product were precipitated within 36 h. The crystals were collected by filtration, washed with cold EtOH (2 mL) and $Et_2O$ (2 × 2 mL), and dried in air. Yield:

65%. Anal. Calcd. (%) for $C_{14}H_{14}N_6ZnBr_2$: C, 34.21; H, 2.88; N, 17.10. Found (%): C, 34.19; H, 3.01; N, 16.78. $\Lambda_M$ (DMSO, 25 °C, $10^{-3}$ M) = 68 S cm$^2$ mol$^{-1}$. $\Lambda_M$ (MeNO$_2$, 25 °C, $10^{-3}$ M) = 12 S cm$^2$ mol$^{-1}$. Selected IR and far-IR data (cm$^{-1}$): 1218s, $\nu_{as}$(NNN); 1135m, $\nu_s$(NNN); 243s, $\nu_{as}$(Zn-Br)$_t$/$\nu_s$(Zn-N); 219s, $\nu_s$(Zn-Br)$_t$; 275s, $\nu_{as}$(Zn-N). Selected Raman peaks (cm$^{-1}$): 1224m, $\nu_{as}$(NNN); 1140s, $\nu_s$(NNN); 246m, $\nu_{as}$(ZnBr)$_t$/$\nu_s$(ZnN); 220s, $\nu_s$(ZnBr)$_t$; 271m, $\nu_{as}$(ZnN). $^1$H NMR ($d_6$-DMSO, $\delta$/ppm; since the spectrum is almost identical with that of Mebta, the proton's numbering scheme follows Figure 1): 8.04 (d, 2H/H4), 7.87 (d, 2H/H7), 7.57 (t, 2H/H6), 7.42 (t, 2H/H5), 4.33 (s, 6H/-CH$_3$). $^{13}$C[$^1$H] NMR ($d_6$-DMSO, $\delta$/ppm; since the spectrum of the complex is identical with that of Mebta, the numbering scheme of carbons is the same adopted in Figure 1): 145.26(C8), 133.48(C9), 127.16(C6), 123.92(C5), 119.07(C4), 110.67(C7), 34.15(C10).

### 3.2.4. [ZnI$_2$(Mebta)$_2$] (**4**)

A solution of ZnI$_2$ (0.080 g, 0.25 mmol) in EtOH (4 mL) was added to a solution of Mebta (0.067 g, 0.50 mmol) in the same solvent (4 mL). The resulting solution was stirred for 30 min, during which time a large amount of a beige solid was precipitated. The solid was filtered and redissolved in MeOH (6 mL). The pale-yellow solution was stored at −10 °C. After 5 d, colorless X-ray quality crystals of the product were grown which were collected by filtration, washed with cold EtOH (2 mL) and Et$_2$O (2 × 2 mL), and dried in air. Yield: 25%. Anal. Calcd. (%) for $C_{14}H_{14}N_6ZnI_2$: C, 28.72; H, 2.41; N, 14.36. Found (%): C, 28.90; H, 2.34; N, 14.51. $\Lambda_M$ (DMSO, 25 °C, $10^{-3}$ M) = 67 S cm$^2$ mol$^{-1}$. $\Lambda_M$ (MeNO$_2$, 25 °C, $10^{-3}$ M) = 14 S cm$^2$ mol$^{-1}$. Selected IR and far-IR data (cm$^{-1}$): 1219s, $\nu_{as}$(NNN); 1132m, $\nu_s$(NNN); 208s, $\nu_{as}$(ZnI)$_t$; 189s, $\nu_s$(ZnI)$_t$; 263m, $\nu_{as}$(ZnN); 248m, $\nu_s$(ZnN). Selected Raman peaks (cm$^{-1}$): 1233w and 1224m, $\nu_{as}$(NNN); 1136s, $\nu_s$(NNN); 208m, $\nu_{as}$(ZnI)$_t$; 182s, $\nu_s$(ZnI)$_t$; 263m, $\nu_{as}$(ZnN); 246m, $\nu_s$(ZnN). $^1$H NMR ($d_6$-DMSO, $\delta$/ppm; same proton numbering scheme as in Figure 1): 8.04 (d, 2H/H4), 7.86 (d, 2H/H7), 7.57 (t, 2H/H6), 7.41 (t, 2H/H5), 4.33 (s, 6H/-CH$_3$). APT $^{13}$C[$^1$H] NMR ($d_6$-DMSO, $\delta$/ppm; same carbon numbering scheme as in Figure 1): Up signals: 145.24(C8), 133.47(C9). Down signals: 127.22(C6), 123.98(C5), 119.08(C4), 110.68(C7), 34.27(C10).

### 3.2.5. tet-[Zn(NO$_3$)$_2$(Mebta)$_2$] (**5**)

To a stirred solution of Zn(NO$_3$)$_2$·4H$_2$O (0.830 g, 3.60 mmol) in EtOH (10 mL) was added dropwise a solution of Mebta (1.000 g, 7.50 mmol) in the same solvent. During the addition, a white solid was precipitated, which was filtered, washed with EtOH (2 × 4 mL) and Et$_2$O (3 × 5 mL), and dried in a vacuum desiccator over silica gel. Yield: 51%. X-ray quality crystals of the product were obtained by performing the same reaction under conditions of high dilution (0.36 mmol of the zinc source and 0.75 mmol of Mebta in 15 mL of EtOH) without stirring during the addition of Mebta (no precipitation took place). The powder and the crystals had identical IR spectra. Anal. Calcd. (%) for $C_{14}H_{14}N_8ZnO_6$: C, 36.89; H, 3.10; N, 24.59. Found (%): C, 37.30; H, 3.00; N, 24.90. $\Lambda_M$ (DMSO, 25 °C, $10^{-3}$ M) = 72 S cm$^2$ mol$^{-1}$. Selected IR and far-IR data (cm$^{-1}$): 1220s, $\nu_{as}$(NNN); 1139m, $\nu_s$(NNN); 1450m, $\nu_5$(B$_2$)[$\nu_{as}$(NO$_2$)]; 1309m, $\nu_1$(A$_1$)[$\nu_s$(NO$_2$)]; 348s, $\nu_{as}$(ZnONO$_2$); 300m, $\nu_s$(ZnONO$_2$); 291s, $\nu_{as}$(ZnN); 263m, $\nu_s$(ZnN). $^1$H NMR ($d_6$-DMSO, $\delta$/ppm; same proton numbering scheme as in Figure 1): 8.03 (d, 2H/H4), 7.87 (d, 2H/H7), 7.58 (t, 2H/H6), 7.42 (t, 2H/H5), 4.33 (s, 6H/-CH$_3$). $^{13}$C[$^1$H] NMR ($d_6$-DMSO, $\delta$/ppm; same carbon numbering scheme as in Figure 1): 145.29(C8), 133.50(C9), 127.14(C6), 123.89(C5), 119.06(C4), 110.60(C7), 34.09(C10).

### 3.2.6. oct-[Zn(NO$_3$)$_2$(Mebta)$_2$] (**6**)

To a stirred solution of Zn(NO$_3$)$_2$·4H$_2$O (0.830 g, 3.60 mmol) in EtOH (10 mL) was added triethyl orthoformate (3 mL). The colorless solution obtained was heated at 55 °C for 45 min (the final volume was ca. 8 mL) and allowed to cool at room temperature. This solution was added dropwise to a stirred solution of Mebta (1.000 g, 7.50 mmol). During the addition, a white solid was precipitated, which was filtered, washed with EtOH

$(2 \times 4$ mL) and $Et_2O$ $(3 \times 5$ mL), and dried in a vacuum desiccator over $P_4O_{10}$. Yield: 62%. X-ray quality crystals of the product were obtained by performing the same reaction under conditions of high dilution (0.36 mmol of the zinc source, 1 mL of triethyl orthoformate, and 0.75 mmol of Mebta in 15 mL of EtOH), without stirring during the addition of Mebta (no precipitation took place). The microcrystalline powder and the crystals had identical IR spectra. Anal. Calcd. (%) for $C_{14}H_{14}N_8ZnO_6$: C, 36.89; H, 3.10; N, 24.59. Found (%): C, 36.64; H, 3.02; N, 24.47. $\Lambda_M$ (DMSO, 25 °C, $10^{-3}$ M) = 81 S cm$^2$ mol$^{-1}$. $\Lambda_M$ (MeNO$_2$, 25 °C, $10^{-3}$ M) = 15 S cm$^2$ mol$^{-1}$. Selected IR and far-IR data (cm$^{-1}$): 1220s, $\nu_{as}$(NNN); 1137m, $\nu_s$(NNN); 1500s, $\nu_1(A_1)[\nu(N=O)]$/aromatic stretch; 1275m, $\nu_5(B_2)[\nu_{as}(NO_2)]$; 326w, 292sh, 274s and 255sh, $\nu(ZnO)_t$; 274s and 242m, $\nu(ZnN)$, the former vibration overlapping with a stretching Zn-O mode. Selected Raman peaks (cm$^{-1}$): 1228m, $\nu_{as}$(NNN); 1141m, $\nu_s$(NNN); 1513s, $\nu_1(A_1)[\nu(N=O)]$/aromatic stretch; 1282m, $\nu_5(B_2)[\nu_{as}(NO_2)]$; 326m, 270s, 260m, $\nu(ZnO)_t$; 270s, 248m, $\nu(ZnN)$, the former overlapping with a stretching Zn-O mode. The $^1$H NMR and $^{13}$C[$^1$H] NMR spectra of the complex in $d_6$-DMSO are identical with the corresponding ones of compound **5**.

### 3.2.7. [Zn(Mebta)$_4$](ClO$_4$)$_2$ (**7**)

To a stirred solution of Zn(ClO$_4$)$_2$·6H$_2$O (0.074 g, 0.20 mmol) in EtOH (2 mL) was slowly added a solution of Mebta (0.107 g, 0.80 mmol) in the same solvent (13 mL). The resulting pale beige solution was stirred for a further 30 min and layered with Et$_2$O (5 mL). Slow mixing gave X-ray quality, colorless crystals of the product after 2 d. The crystals were collected by filtration, washed with cold EtOH (1 mL) and Et$_2$O ($2 \times 2$ mL), and dried in air. Yield: 45%. Anal. Calcd. (%) for $C_{28}H_{28}N_{12}ZnCl_2O_8$: C, 42.20; H, 3.55; N, 21.10. Found (%): C, 42.60; H, 3.25; N, 21.47. $\Lambda_M$ (DMSO, 25 °C, $10^{-3}$ M) = 71 S cm$^2$ mol$^{-1}$. Selected IR and far-IR data (cm$^{-1}$): 1218m, $\nu_{as}$(NNN); 1144s, $\nu_s$(NNN); 1112s, $\nu_3(F_2)[\nu_d(ClO)]$; 630s, $\nu_4(F_2)[\delta_d(OClO)]$; 288s, $\nu(ZnN)[T_2]$. $^1$H NMR ($d_6$-DMSO, $\delta$/ppm; since the spectrum of the complex is practically identical with that of free Mebta, the numbering schemes of protons follows Figure 1): 8.06 (d, 4H/H4), 7.87 (d, 4H/H7), 7.55 (t, 4H/H6), 7.46 (t, 4H/H5), 4.32 (s, 12H/-CH$_3$). $^{13}$C[$^1$H] NMR ($d_6$-DMSO, $\delta$/ppm; same carbons' numbering scheme as in Figure 1): 145.27(C8), 133.51(C9), 127.27(C6), 123.96(C5), 119.13(C4), 110.68(C7), 34.23(C10).

### 3.2.8. [Zn(Mebta)$_4$](PF$_6$)$_2$ (**8**)

To a stirred solution of Mebta (0.107 g, 0.80 mmol) in EtOH (2 mL) was added a solution of NH$_4$(PF$_6$) (0.130 g, 0.80 mmol) in EtOH (8 mL) and the resulting solution was stirred for a further 10 min. This solution was combined with a solution of Zn(ClO$_4$)$_2$·6H$_2$O (0.074 g, 0.20 mmol) in the same solvent (2 mL). The final reaction solution was layered with n-hexane (5 mL). Slow mixing gave medium quality crystals after 2 d. The crystals were collected by filtration, washed with EtOH (2 mL) and Et$_2$O ($2 \times 2$ mL), and dried in air. Yield: 59%. Anal. Calcd. (%) for $C_{28}H_{28}N_{12}ZnP_2F_{12}$: C, 37.87; H, 3.18; N, 18.93. Found (%): C, 38.27; H, 2.99; N, 19.13. $\Lambda_M$ (DMSO, 25 °C, $10^{-3}$ M) = 79 S cm$^2$ mol$^{-1}$. IR and far-IR data (cm$^{-1}$): 1226m, $\nu_{as}$(NNN); 1170w, $\nu_s$(NNN); 834s, $\nu_3(F_{1u})[\nu(PF)]$; 558m, $\nu_4(F_{1u})[\delta(FPF)]$; 287s, $\nu(ZnN)[T_2]$. $^1$H NMR ($d_6$-DMSO, $\delta$/ppm; same protons' numbering scheme as in Figure 1): 8.04 (d, 4H/H4), 7.87 (d, 4H/H7), 7.56 (t, 4H/H6), 7.45 (t, 4H/H5), 4.33 (s, 12H/-CH$_3$). $^{13}$C[$^1$H] NMR ($d_6$-DMSO, $\delta$/ppm; same numbering scheme of carbons as in Figure 1): 145.28(C8), 133.52(C9), 127.27(C6), 123.95(C5), 119.14(C4), 110.66(C7), 34.25(C10).

### 3.3. Single-Crystal X-ray Crystallography

Colorless crystals of **3**, **4**, and **7** were taken directly from the mother liquor and immediately cooled to 160 K ($-113$ °C). Diffraction data were collected on a Rigaku R-Axis Image Plate diffractometer using graphite-monochromated Cu K$_\alpha$ radiation. Data collection ($\omega$-scans) and processing (cell refinement, data reduction, and empirical/numerical correction) were performed using the CrystalClear program package [68]. A colorless crystal of **6** was mounted in air and covered with epoxy glue. Diffraction measurements were performed on a Crystal Logic Dual Goniometer diffractometer using graphite-monochromated Mo

$K_\alpha$ radiation. Unit cell dimensions were determined by using the angular settings of 25 automatically centered reflections in the range $11 < 2\theta < 23°$. A colorless crystal of **5** was also mounted in air and covered with epoxy glue. Diffraction data were collected on a P2$_1$ Nicolet diffractometer upgraded by Crystal Logic using graphite-monochromated Cu $K_\alpha$ radiation. Unit cell dimensions were determined by using the angular settings of 25 automatically centered reflections in the range $22 < 2\theta < 54°$. Intensity data for **5** and **6** were recorded using a $\theta$-$2\theta$ scan; three standard reflections, monitored every 97 reflections, showed less than 3% variation and no decay. Lorentz and polarization corrections were applied using Crystal Logic software.

The structures were solved by direct methods using SHELXS, ver. 2013/1 [69] and refined by full-matrix least squares techniques on F$^2$ with SHELXL, ver. 2014/6 [70]. Plots of the structures were drawn using the Diamond 3 program package [71]. For the structures of **3**, **5**, **6**, and **7**, H atoms were located by difference maps and refined isotropically; all non-H atoms were refined anisotropically. For **4**, H atoms were introduced at calculated positions as riding on their corresponding bonded atoms; all non-H atoms were refined anisotropically. In the structure of **7**, there are three crystallographically independent perchlorates in the asymmetric unit; two sit on a 2-fold axis of symmetry and the third sits on a general position. The former two each contribute half ClO$_4$$^-$ ions in the asymmetric unit of the cell and the third contributes one. Crystallographic data can be found in Tables S7 and S8; more details are in the CIF files. Some crystallographic data for **8**, whose structure is of rather poor quality, have already been mentioned in part 2.2 ("Description of Structures").

Crystallographic data were deposited with the Cambridge Crystallographic Data Center, Nos. 2286175-2286179. Copies of the data can be obtained free of charge upon application to CCDC, 12 Union Road, Cambridge, CB2 1EZ: Telephone: +(44)-1223-336033; E-mail: deposit@ccdc.ac.uk, or via https://www.ccdc.cam.ac.uk/structures/ (accessed on 20 June 2023).

*3.4. Computational Details*

The MP2 [72–76] ab initio methodology was applied on the free Mebta molecule with the 6+31+G(d) basis set in EtOH solution (all its complexes in this work were prepared in this solvent), using the polarized continuum model (PCM) [77] for the treatment of the solvent effects. The DFT methodology with ωw97XD functional [78–88] (containing empirical dispersion terms and long-range corrections and thus providing good descriptions of the reaction profiles, including geometries, heats of reactions, and barrier heights) was used for complexes **5** and **6**, as implemented in the Gaussian09 programs suite [83]. Two different calculations were performed for each complex. In the first denoted as BSI, which does not include relativistic effects for the Zn atom, the 6-311+G(d,p) basis set was used for all atoms. In the second one denoted as BSII, which includes relativistic effects for the Zn atom, the Def2-TZVP [84–88] basis set was used for all atoms. Full geometry optimizations with no symmetry constraints were performed for the compounds. Harmonic frequencies were computed at the same level of theory, and the nature of the stationary points was determined in each case according to the number of the negative eigenvalues of the Hessian matrix (zero and one imaginary frequencies for the minima and transition states, respectively).

**4. Concluding Comments and Perspectives**

In this work, we reported the employment of Mebta in reactions with Zn(II) sources, which has resulted in the isolation of eight complexes. Thus, the empty position of Zn(II) in the "Periodic Table" of Mebta has now been filled in. The most salient features of this work are: (a) Mebta can form anionic (**2**), neutral (**1**, **3**–**6**), and cationic (**7**, **8**) complexes with interesting molecular structures; the anionic complex (MebtaH)$_2$[ZnCl$_4$] (**2**) undergoes solid-state decomposition yielding [ZnCl$_2$(Mebta)$_2$] (**1**) reflecting the high acidic character of the organic cation. (b) The reactions of Zn(NO$_3$)$_2$·4H$_2$O and Mebta have provided access

to two isostoichiometric complexes, depending on the reaction conditions, with different stereochemistries at Zn$^{II}$; this difference arises from the monodentate vs. bidentate coordination of the nitrato ligands. (c) The Mebta ligands favor the formation of 2D layers through π–π interactions in the structures of **3**, **5**, and **7**; with the aid of H bonds developed among species belonging to neighboring layers, the 3D architecture of these compounds is built. Small changes in the arrangement of ligands observed in the structures of **4** and **6** result in the formation of chains through π–π interactions, and H-bonding interactions among molecules belonging to neighboring chains contribute to the construction of the architecture of these complexes. (d) The thermal decomposition of [ZnX$_2$(Mebta)$_2$] (X = Cl, **1**; X = Br, **3**) leads to stable intermediates (**1a** and **3a**, respectively) with the 1:1 stoichiometry, i.e., ZnX$_2$(Mebta), which are probably polymeric tetrahedral complexes containing monodentate Mebta, and both terminal and bridging halide groups; and (e) Liquid-phase (Mebta) and gas-phase (**5**, **6**) DFT calculations satisfactorily explain the preference of Mebta for N3-coordination (and not N2-coordination) and the existence of **5** and **6** which seem to have a comparable thermodynamic stability.

With the knowledge and experience obtained in this work, our future research plans are directed, among others, to: (1) The use of carboxylate-containing Zn(II) starting materials in Mebta coordination chemistry; preliminary results show the preparation of coordination clusters, e.g., [Zn$_3$(O$_2$CR)$_6$(Mebta)$_2$], as a consequence of the great bridging capability of the carboxylate groups; (2) The study of the Cd(II)/Mebta general reaction system with the goal to establish similarities and differences in the behavior of Zn(II) and Cd(II), which are both group 12 metals; and (3) The extension of the Mebta reactions with other, except Fe(III) [51], trivalent [e.g., Ga(III), In(III), and Tl(III)] and tetravalent [e.g., Sn(IV) and Pb(IV)] metals in order to complete other empty positions in the "Periodic Table" of Mebta.

**Supplementary Materials:** The following supporting information can be downloaded at: https://www.mdpi.com/article/10.3390/inorganics11090356/s1. Figure S1: Partially labeled ball and stick representation of the structure of the molecule [ZnBr$_2$(Mebta)$_2$] that is present in **3**; Figure S2: Partially labeled ball and stick representation of the structure of the molecule [ZnI$_2$(Mebta)$_2$] that is present in **4**; Figure S3: Supramolecular characteristics of complex **3**; Figure S4: Partially labeled ball and stick representation of the structure of the molecule tet-[Zn(NO$_3$)$_2$(Mebta)$_2$] that is present in **5**; Figure S5: Supramolecular characteristics of complex **5**; Figure S6: Partially labeled ball and stick representation of the structure of the molecule oct-[Zn(NO$_3$)$_2$(Mebta)$_2$] that is present in **6**; Figure S7: Supramolecular characteristics of complex **6**; Figure S8: Partially labeled ball and stick representation of the structure of the cation that is present in **7**; Figure S9: The $^{13}$C[$^1$H] NMR spectrum of Mebta in $d_6$-DMSO; Figure S10: The $^{13}$C[$^1$H] NMR spectrum of **1** in $d_6$-DMSO; Figure S11: The APT $^{13}$C[$^1$H] NMR spectrum of **4** in $d_6$-DMSO; Figure S12: The IR spectrum of **1**; Figure S13: The IR spectrum of **7**; Figure S14: The IR spectrum of **8**; Figure S15: The FT far-IR spectrum of **2**; Figure S16: The FT far-IR spectrum of **6**; Figure S17: X-ray and DFT calculated gas-phase structure of **5**; Figure S18: X-ray and DFT calculated gas-phase structure of **6**. Table S1: Experimental and computationally derived bond lengths and angles of **5**; Table S2: Experimental and computationally derived bond lengths and angles of **6**; Table S3: Calculated liquid-phase Cartesian coordinates of Mebta; Table S4: Calculated [wB97XD/6-311+G** protocol (BSI)] gas-phase Cartesian coordinates of **5** and **6**; Table S5: Gas-phase Cartesian coordinates of **5** and **6**, calculated by the wB97XD/Def2tzvp computational protocol (BSII); Table S6: Sum of electronic and thermal free energies (a.u.) of **5** and **6**; Table S7: Crystallographic data and refinement parameters for **3** and **4**; Table S8: Crystallographic data and refinement parameters for **5**–**7**. The CIF and the check CIF output files are included in the Supplementary Materials.

**Author Contributions:** C.S. and E.B. contributed towards the synthesis, crystallization, and characterization of the complexes. J.C.P. synthesized some complexes and performed an extensive literature search; he also recorded the far-IR and $^{13}$C[$^1$H] NMR spectra. M.M.S. contributed towards theoretical calculations. C.P.R. and V.P. solved the single-crystal X-ray structures; the latter also studied the supramolecular features of the reported structures and wrote the relevant part of the paper. E.G.B. performed detailed DFT studies and wrote the relevant part of the paper. J.C.P. and S.P.P. coordinated the research; the latter wrote the paper based on the detailed reports of the collaborators. All the au-

thors exchanged opinions concerning the progress of the project and commented on the writing of the manuscript at all steps. All authors have read and agreed to the published version of the manuscript.

**Funding:** C.P.R. and V.P. would like to thank the Special Account of the NCSR "Demokritos" for financial support concerning the operation of the X-ray facilities at INN through the internal program entitled "Structural Study and Characterization of Crystalline Materials" (NCSR "Demokritos", ELKE ≠ 10813).

**Data Availability Statement:** Not applicable.

**Acknowledgments:** S.P.P. would like to thank Professor Spyros Yiannopoulos for the purchase of a quantity of Mebta.

**Conflicts of Interest:** The authors declare no conflict of interest.

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
