# Peer review of "The “Periodic Table” of 1-methylbenzotriazole: Zinc(II) Complexes"

_inorganics, doi:10.3390/inorganics11090356_

Round 1

Reviewer 1 Report

In the manuscript entitled: “The “Periodic Table” of 1-methylbenzotriazole: Zinc(II) Complexes” the authors describe the synthesis, single crystal X-ray structure determination, spectroscopic characterization, and DFT calculation of zinc(II) complexes with 1-methylbenzotriazole (Mebta). The theme of the manuscript is interesting with clear objectives of the work. The description of the compounds is systematic and gave a clear contribution to chemistry. Nevertheless, there are a few problems that need correction to improve the quality of the manuscript.

Line 165: Instead of “Figures 1, 2, S1 and S2” Figures 2, 3, S1 and S2 should stay.

Lines 175-179: Knowing the azoles, the decrease of the double bond character of the triazole ring because of the coordination was expected in my opinion. It should be mentioned in the discussion.

Table 3 should be cited in the description of the molecular structure of complex 7 not only at the beginning of that section.

Line 338-341: The presented values of molar conductivity in DMSO better fit to electrolytes type 1:1 than 2:1 regarding cited paper [Geary, W.J. The use of conductivity measurements in organic solvents for the characterization of coordination compounds. 980 Coord. Chem. Rev. 1971, 7, 81-122.] This part should be proved and/or some explanation given.

Line 384: For Complex oct-[Zn(NO3)2(Mebta)2] “the final plateau >250 °C corresponds to ZnO”. Is this decomposition product analyzed and proved its identity or only supposed?

Line 423: All far-IR spectra are discussed but only two are presented in the manuscript. It would be very useful for the reader and more representative to give in one figure all discussed spectra.

Lines 462-463: Citation 72 should be replaced in the next line after “vibrations”.

Section 3.1. should be formatted as all other sections.

The figures should be presented in higher resolution.

Following the aforementioned, I recommend the acceptance of the manuscript after minor revision.

Author Response

We are grateful to Reviewer 1 for her/his time to study the ms, and provide us with valuable comments and suggestions. She/he examined the ms line-by-line and all the revision comments were clear and well organized. Her/his contribution to improve the quality of the ms is great!

We are pleased to inform the reviewer that we have addressed almost all her/his revision points/comments/suggestions raised. For convenience and easy communication, we include each comment before our answer.

 “Line 165: Instead of “Figures 1, 2, S1 and S2” Figures 2, 3, S1 and S2 should stay.”

   This correction has been performed.

“Lines 175-179: Knowing the azoles, the decrease of the double bond character of the triazole ring because of the coordination was expected in my opinion. It should be mentioned in the discussion.”

The comment is absolutely correct. We have added a sentence highlighting the decrease of the double bond character of the triazole ring due to coordination.

“Table 3 should be cited in the description of the molecular structure of complex 7 not only at the beginning of that section.”

This correct note has been taken into account.

“Line 338-341: The presented values of molar conductivity in DMSO better fit to electrolytes type 1:1 than 2:1 regarding cited paper [Geary, W.J. The use of conductivity measurements in organic solvents for the characterization of coordination compounds. 980 Coord. Chem. Rev. 1971, 7, 81-122.] This part should be proved and/or some explanation given.”

The comment is correct. As a matter of fact, the molar conductivity values for some of our complexes in DMSO are intermediate between those reported for 1:1 and 1:2 electrolytes. We have added an alternative explanation for this in the appropriate place of part 2.3, as requested by the reviewer. With our more than 45-year (!) experience on the use of molar conductivity measurements in Coordination Chemistry, we do believe that values around 70 S cm2 mol-1, as observed in complexes 1, 3, 4 and 5, are indicative of 1:2 electrolytes in DMSO. Such values have been often reported in the literature (see ref. [57] and the helpful paper sent to us by The Academic Editor). This is also proven by the value for the ionic complex 7 (71 S cm2 mol-1), which is –no doubt- an 1:2 electrolyte.

“Line 384: For Complex oct-[Zn(NO3)2(Mebta)2] “the final plateau >250 °C corresponds to ZnO”. Is this decomposition product analyzed and proved its identity or only supposed?”

The question is scientifically correct and logical. The assumption was based only on mass loss calculations. We have added a new sentence clarifying this point.

“Line 423: All far-IR spectra are discussed but only two are presented in the manuscript. It would be very useful for the reader and more representative to give in one figure all discussed spectra.”

Although we tried, the incorporation of all the FT far-IR spectra in one figure decreased its quality. This is partly due to the fact that the instrument used (Bruker IFS 113v) is now out-of-order and its software no available to us. Thus, we ask your and Reviewer’s 1 indulgence to leave the separate Figures 9 and 10 in the main text of the revised ms. However, in order to show that we respect the reviewer’s opinion, we have added the FT far-IR complexes of two more complexes (compounds 2 and 6) in the now revised “Supplementary Materials” section (Figures S15 and S16); the previous Figures S15 and S16 have now numbered Figures S17 and S18.

“Lines 462-463: Citation 72 should be replaced in the next line after “vibrations”.”

We have performed this correction.

“Section 3.1. should be formatted as all other sections.”

We had noticed this discrepancy during the initial submission of the ms. We are not responsible for this. This is a technical problem in the submission system of “Inorganics”. We can not resolve this problem.

“The figures should be presented in higher resolution.”

We have improved the resolution of many figures of the main revised ms, namely Figures 2, 3, 4 (both a and b), 5, 6, 7 and 8.

Reviewer 2 Report

Psycharis, Perlepes et al report the synthesis and characterization of a new series of Zn(II) complexes having 1-methylbenzotriazole (Mebta) as ligand.

The manuscript is original, easy to read, well written, detailed and complete therefore, in my opinion, it is suitable for publication on Inorganics in actual form.

Author Response

We are grateful to Reviewer 2 for her/his time to study the ms and the positive proposal for acceptance without corrections and revisions.